# The effects of social feedback on private opinions. Empirical evidence from the laboratory

**Marcel Sarközi**[1☯]*, **Stephanie Jütersonke**[2☯], **Sven Banisch**[3], **Stephan Poppe**[1], **Roger Berger**[1]

**1** Institute of Sociology, Leipzig University, Leipzig, Germany, **2** Institute of Sociology, University of Tübingen, Tübingen, Germany, **3** Max Planck Institute for Mathematics in the Sciences, Leipzig, Germany

☯ These authors contributed equally to this work.
* marcel.sarkoezi@posteo.de

**Data Availability Statement:** The dataset as well as the Stata do-file used for the analyses are available from the figshare project repository. DOI: https://doi.org/10.6084/m9.figshare.16595378.v1.

## Abstract

The question of how people change their opinions through social interactions has been on the agenda of social scientific research for many decades. Now that the Internet has led to an ever greater interconnectedness and new forms of exchange that seem to go hand in hand with increasing political polarization, it is once again gaining in relevance. Most recently, the field of opinion dynamics has been complemented by social feedback theory, which explains opinion polarization phenomena by means of a reinforcement learning mechanism. According to the assumptions, individuals not only evaluate the opinion alternatives available to them based on the social feedback received as a result of expressing an opinion within a certain social environment. Rather, they also internalize the expected and thus rewarded opinion to the point where it becomes their actual private opinion. In order to put the implications of social feedback theory to a test, we conducted a randomized controlled laboratory experiment. The study combined preceding and follow-up opinion measurements via online surveys with a laboratory treatment. Social feedback was found to have longer-term effects on private opinions, even when received in an anonymous and sanction free setting. Interestingly and contrary to our expectations, however, it was the mixture of supportive and rejective social feedback that resulted in the strongest influence. In addition, we observed a high degree of opinion volatility, highlighting the need for further research to help identify additional internal and external factors that might influence whether and how social feedback affects private opinions.

## Introduction

The same epoch that has witnessed the unprecedented technical extension of communication has also brought into existence the deliberate manipulation of opinion and the "engineering of consent". There are many good reasons why, as citizens and as scientists, we

**Funding:** This project has received funding from the European Union's "H2020 EXCELLENT SCIENCE - Future and Emerging Technologies (FET)" programme under grant agreement number 732942. The funders had no role in study design, data collection and analysis, decision to publish, or preparation of the manuscript.

**Competing interests:** The authors have declared that no competing interests exist.

should be concerned with studying the ways in which human beings form their opinions and the role that social conditions play.

([1], p. 31)

Solomon Asch had good reason to encourage fellow social scientists to join him in his attempt to understand why and how human beings develop and change their opinions. His famous group experiments on conformity revealed how willing participants were to conform to group behavior and abandon their own views no matter how objectively skewed the majority opinion was [1, 2]. Confronted with the simple task of visually comparing the length of lines, a remarkable number of participants adjusted their estimates in the direction of those given by the other subjects in the room, who had secretly been instructed to answer erroneously. The results, highlighting the potential impact of social influence on opinions, led to a question which has inspired many and will be guiding the paper at hand: "Exactly what is the effect of the opinions of others on our own?" ([1], p. 31).

The extent to which Asch's conformity experiments answer this question is, however, limited. This becomes apparent by the reasoning that yielding participants offered when asked about their change of heart. Only very few fully adapted and simultaneously claimed not to have been aware of the influence of the majority opinion on their own. Most of the yielding subjects admitted having deliberately adopted the unanimous estimate of their group members. The subsequently reported reasons for their individual decisions varied greatly: While some were honestly convinced that their own perception was flawed, others confessed that they conformed due to fear of judgment.

Evidence from another series of classic experiments on social influence reveals that Asch's participants had good reason to believe that non-adjustment might expose them to undesired outcomes. When observing influencing behavior in groups, George Homans discovered that participants insisting on minority positions were more likely to face social exclusion or punishment from those they disagreed with [3]. These findings corroborated his postulate that individuals find value in other people agreeing with them and fit well in his theoretical frame-work of social exchange which provides a pathway to understanding social influence on opinion change. Central to the argument of social exchange theory is the assumption that individuals, when interacting with each other, award the activities of their fellows with either reward or punishment. Rewards, or reinforcers as Homans calls them, make a certain behavior more likely, while the withdrawal of reinforcers or the punishment of behavior leads to a decline in likelihood. In situations of social exchange, intangible goods such as sentiments can serve as either reinforcers or punishers. Among these, social approval is of particular interest for it is possible to encourage a broad variety of human activities by rewarding them with this kind of affirmation. In combining these thoughts, Homans concludes that people give social approval to others that have given them an activity they value. This reinforcement in turn makes it more likely that the others will repeat the action in question. As similarity in behavior is one feature that individuals find valuable, open disagreement thus can be expected to result in punishment to the point of exclusion.

The assumption that individuals adjust their public opinion expression due to fear of social isolation is also central for the spiral of silence theory put forth by Elisabeth Noelle-Neumann [4]. According to her conceptualization, people are in a constant and mostly subconscious state of monitoring opinion sources. When deciding on whether to speak out or to remain silent within a certain social environment they let their subjective impression of public majority opinion guide them in figuring out which opinions are prevalent and therefore safe to express. According to Noelle-Neumann, learning from knowledgeable others can be identified as an important cause for changes in publicly expressed opinions.

Nowadays, this idea seems worthy to reconsider as the vast spread of Internet usage has enabled people to observe and exchange information as well as opinions at an unprecedented rate; a development which has raised hopes and fears alike among scholars of opinion dynamics. The optimists on the one side expect the emergence of a new online public sphere in which citizens will experience exposure to political discussion and cross-cutting information, which has been known for it's moderating effect on political dissent and stimulating impact on political participation [5–8]. The skeptics on the other side fear that individuals will either refrain from exchange with those that think differently or embrace more and more opposite standpoints when interacting with them [9, 10]. They expect online communication to lead to an increase in opinion polarization, a state that is feared by many for its undermining consequences on social and political stability [11].

## Public expression and private opinion

The approaches and findings outlined above share a more or less explicit focus on publicly expressed opinions. But social scientists have long been aware of the fact that "the opinions a person expresses publicly may diverge in varying degrees from those which he holds in private" ([12], p. 427). This raises the question of what we can actually learn about the effects of social influence from simply observing adjustment of a publicly expressed opinion which might be motivated by group pressure.

It was only a few years after Asch that Kelman called for a better understanding of the conditions under which opinions are formed and changed, adopted and abandoned, and likely to be expressed [13]. His own theoretical reflections started out from a rather basic but useful distinction between two potential outcomes of social influence processes: public conformity and private acceptance. While the former is characterized through a verbal and situational adaptation that is not necessarily preceded by a conversion of the individual's beliefs, the latter represents a more general and enduring internal change of opinion. Kelman expatiated three processes by which people adopt opinions in the wake of social influence: compliance, identification, and internalization. The latter, in particular, is associated with private acceptance, as it occurs when a person finds the content of an influence intrinsically rewarding and "inherently conductive to the maximization of his values" ([13], p. 65). Publicly expressed opinions, however, that are due to compliance with surrounding others are prone to vary with the situational circumstances [14]. The revealing finding of how, as in Asch's experiments [1, 2], vastly different motivations on the individual level can lead to the same type of observed behavioral outcome exemplifies the extent to which the factual effects of social influence can go by unacknowledged when simply measuring differences in public expression. Capturing the changes that occur within the individual is thus essential for making predictions of subsequent behavior, or when trying to understand whether and under what conditions opinions form, persist, change, and translate into action. Therefore, when concerned with long-term changes in internal convictions and subsequent behavior, researchers have to turn their attention towards analyzing the changes in *private opinions*, the ones that individuals actually hold.

## Reinforcement learning from social feedback

The paper at hand aims to connect these well known challenges of opinion research with the new ones that online communication has brought about. In doing so, we find findings familiar from opinion dynamics research also in the emerging patterns of online communication. In the former, researchers are struggling to explain the persistence of opinion differences between interacting individuals which is at odds with both, empirical evidence from social psychology that shows that people tend to assimilate their opinions to those of people they interact with as

well as classical social influence models that predict opinion convergence among interacting individuals in the long run [15–17].

In order to explain these phenomena of opinion divergence, a number of opinion exchange models and social influence mechanisms have been suggested. One of the most recent attempts was the introduction of a reinforcement learning mechanism that is based on *social feedback* [18, 19]. The affective experience-based approach centers around Homans' idea that individuals repeat behavior that is rewarded by others [3] and argues along the same lines as Noelle-Neumann [4]: When interacting and expressing opinions within a given social environment, individuals receive social feedback in return. Since human beings generally are sensitive to approval and disapproval [3], they subsequently use the judgments they receive to develop an idea about the prevalent opinions in their social neighborhood and internally evaluate available opinion alternatives. The positive experience of supportive social feedback is intrinsically rewarding and thus leads to a strengthening in the internal attachment to the expressed opinion. Negative, i.e. rejective social feedback, correspondingly, results in decreased adherence. Within the course of such a reward-driven reinforcement process, individuals not only reevaluate the opinions expressed and learn which are safe to express in their neighborhood. They also internalize the expected opinion and integrate it with their existing values until it "gradually becomes independent of the external source", as Kelman ([13], p. 66) put it, as well as independent of the possibility of observation. Reinforcement learning from social feedback is therefore expected to go beyond effects of mere public conformity. It rather leads to actual changes in internal convictions and, ultimately, in private opinions.

## Research questions and hypotheses

Yet up to now, there is no empirical validation for the presumed effects of social feedback. Echoing the ideas and questions that have driven research on social influence and opinion dynamics alike, our empirical approach is motivated by the following research questions:

> *Does social feedback yield any relevant effect on private opinions? And furthermore, does any potentially resulting change of opinion correspond to whether the social feedback was supportive or rejective in its nature?*

From the theoretical groundwork that was presented in the previous section, and in order to provide first empirically supported answers for our research questions, we derived and experimentally tested the following basic hypotheses.

**H1** If social feedback is perceived as *positive*, the private opinion is strengthened and shifts further in the direction of the original position.

**H2** If social feedback is perceived as *negative*, the private opinion is weakened and shifts in opposite direction of the original position.

Suppose a private opinion is represented by a person's position on a symmetrical scale ranging from complete disagreement to complete agreement on an issue, with the neutral position in the middle. In that case our hypotheses imply the following empirical consequences: If **H1** is correct and a person holds (*A*) a disagreeing private opinion of a certain extent, receiving positive, i.e. supportive social feedback leads to more disagreement. Analogously, if the same person holds (*B*) an agreeing private opinion, she would agree even more following the positive feedback. In contrast, if **H2** is correct a person with (*A*) a disagreeing private opinion responds to negative, i.e. rejective social feedback by moving towards the anticipated position of the person giving the feedback and thus taking a more agreeing position than before. In the

case of an originally (*B*) agreeing private opinion, on the other hand, it is expected that the extent of agreement will be reduced.

**H3** If social feedback occurs in a *balanced mixture*, the effects of positive and negative judgments cancel each other out and the private opinion will not be affected.

Since it is assumed that neither positive nor negative judgments are superior to their respective counterpart type of social feedback in terms of influential strength, we additionally propose a third hypothesis **H3**, according to which an equal amount of positive and negative judgments does not result in a change of opinion. Consequently, individuals who receive such mixed feedback from their social environment are expected to act just as they would have in the counterfactual state, in which they had received no social feedback at all.

## Experimental study

### Design and sample

In order to test these hypotheses we conducted a randomized controlled laboratory experiment. The study consisted of three parts which were carried out at three consecutive points in time. While participants' initial (pretest) and final (posttest) private opinions were measured through anonymous standardized online surveys at times $t_1$ and $t_3$, the core part of the study, which was the attempted manipulation of a particular private opinion through exposure to social feedback, took place in our laboratory at $t_2$ (see Fig 1).

Randomization was realized in two steps, as we first randomly assigned participants to either the control or the treatment group and in a second step randomly assigned treatment group participants to one of three different experimental conditions. Subjects had registered as voluntary test persons for experimental research at the Leipzig Experimental Laboratory for Social Sciences (LEx). Within the first e-mail, subjects were invited to participate in a multi-part study, which they could start off by completing the linked online survey. A 20 Euro compensation fee was granted to volunteers who would participate in at least two parts of the study. Wording left the overall number and intention of the study parts open, so that members of neither control nor treatment group knew about the existence of the other group or a variation in procedure. The actual treatment process started after the first online survey was completed, with treatment group members taking part in one of the 18 laboratory sessions. The second online survey concluded the experimental process. A total of 270 people participated in all parts of the study to which they were invited. Overall, the study spanned over a period of 23 days. Since time spans between participation in the lab and the posttest could range from zero to 14 days, the study design allows the observance of durable and integrated treatment effects on actual private opinions which go beyond mere situational and spontaneous shifts in publicly given responses.

To further ensure this crucial study feature, the explicit assurance of permanent anonymity played a prominent role throughout the whole study process. Within the online surveys it increased the likelihood of undistorted and honest responses by the participants, and, therefore, ensured the assessment of what we refer to as private opinions. In the laboratory, the combination of an anonymous setting with a computer-assisted treatment process, which allowed to control all of the content to which subjects were exposed to, made sure that potential social feedback effects would not be obscured by characteristics of other subjects present. This was of particular importance, as it has long been established that socially significant characteristics affect the ability of individuals to assert themselves and convince others in groups [20–24]. In social influence processes individuals thus may show compliance or adaptation because they regard the influencer as a credible source of information or because they want to meet anticipated expectations [13]. To arrive at respective assessments, however, individuals must have access to relevant

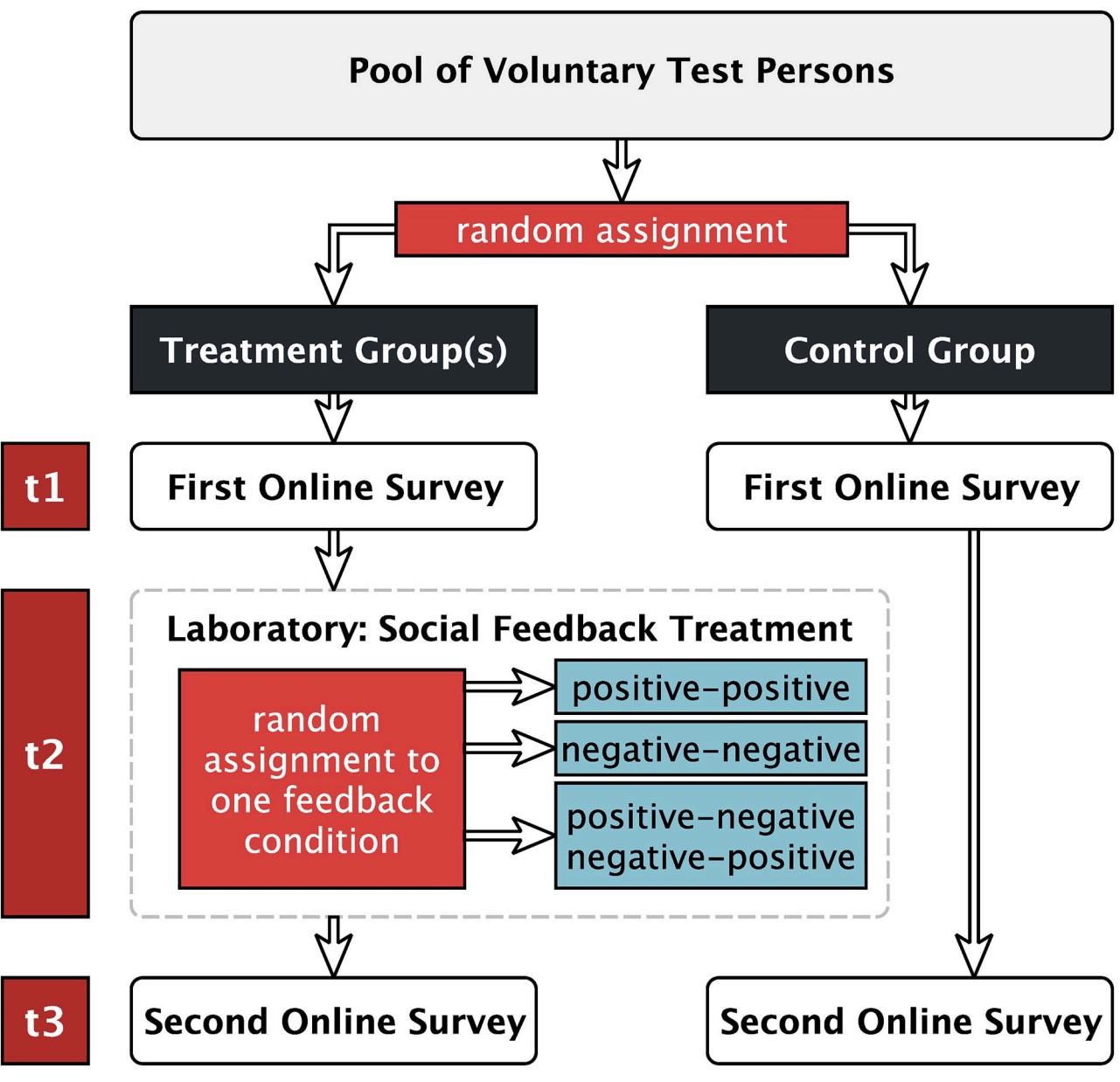

**Fig 1. The course of the study.**

characteristics of the influencing group or person. In fact, removing social cues like race, gender, age, organizational position, etc. has been shown to cause an increased equality of influence across status and expertise in groups [25]. Consequently, by excluding this type of influential variables the presented experimental design ensured high internal validity while also resembling the rather anonymous setting of online communication. It is, therefore, expected to allow for an investigation of the isolated effects of social feedback on private opinions.

### Measurement of private opinions

Measuring the impact of social feedback on private opinions required a study topic that was neither subject to general social consensus nor strongly polarized. It also had to be common as

well as emotionally charged enough for subjects to care about other participants' judgments. Taking these considerations into account, we chose to construct the experiment around an alleged study on "the relationship between humans and animals"; a topic that has been subject to a widespread, morally loaded, and rather emotional public debate in recent years.

Private opinions were measured using the Speciesism Scale, an instrument that intends to capture humans' discriminatory inclinations towards members of other species [26]. The scale was at the core of both online questionnaires at $t_1$ and $t_3$, which were realized by means of SoSci Survey [27] and were completed at a time as well as in an environment of the individuals' choice. For each of the 23 items of the Speciesism Scale participants were asked to indicate the extent of their disagreement or agreement on a slider ranging from 0 (complete disagreement) to 100 (complete agreement). Since we were aiming to use an outcome variable that was as evenly distributed as possible at the outset of the study, the opinion distributions of each item, as measured at $t_1$, served as a basis for choosing the target item, i.e. the private opinion that participants would receive social feedback for after expressing their opinion during a laboratory session. We decided to use the following target item:

> "The killing and eating of animals is part of human nature."

## Laboratory treatment process

The laboratory sessions were conducted in our computer pool at time $t_2$. Participants were seated at individual work stations, which were isolated by three-sided walls. On average, 12 participants were present in each lab session. They were told that the computer would divide all attending participants randomly into groups of three, who would then interact in three consecutive rounds of opinion exchange. It was stressed that the identity of group members would never be disclosed. The opinion exchange process, as it was presented to the participants, would start with one of the members of each group giving an opinion on a particular statement which would in the following be presented to and judged by the two other group members. Their judgments would then be presented to the opinion giving person only. Reportedly, the entire process was to be repeated three times until each one of them had stated an opinion and judged the opinions of both other group members once.

However, in reality there was no group formation process and no interaction between participants whatsoever. Instead, and unbeknownst to the attending individuals, each one of them went through the same process in exactly the same order, stating their opinion on the target item at the beginning of the first round and receiving two pre-formulated feedback statements at the end of it. All subjects were asked to indicate their opinion on the target item on a 0 to 100 slider in the same way as they had done in the first online survey. This time, though, they were told that the opinion would be presented to the other two group members afterwards. Whilst waiting for the alleged judgments of those fictional group members, participants had the opportunity to further explain their position on a waiting screen, with the sounds of their keyboards giving the impression that social feedback from others was being written down at the same time. After five minutes had passed, the anonymous social feedback statements of the alleged other group members were presented automatically. All of those pre-formulated sentences were completely randomly and independently assigned by the computer. This random assignment of social feedback thus has to be considered as the experimental treatment and second randomization step of the presented study. The second and third round of each laboratory session were staged in order to maintain the illusion that the experiment was conducted as

presented in the beginning. The laboratory feedback process was created and carried out by means of z-Tree [28].

Overall, 20 different social feedback statements were distributed among which ten were positive (supportive) and ten were negative (rejective); see S1 Table for a complete list. Participants were simultaneously exposed to two judgments whereby upholding the illusion that they were interacting with two other group members, namely "A" and "C". The feedback statement assignment resulted in three different treatment groups with participants receiving either two positive (positive feedback condition), two negative (negative feedback condition) or a combination of one positive and one negative feedback statement in varying order (mixed feedback condition). While having varying combinations of social feedback statements subsumed under the labels of each treatment condition could be argued to have introduced treatment heterogeneity, this procedure was necessary as it reduced the risk of participants seeing through the deception afterwards. At the end of the second online survey, participants were asked to evaluate to what degree they perceived each of the 20 statements used in the lab as positive or negative. We found that only in four single cases a participant's perception of a certain statement differed from our assessments.

## Ethics statement

The basic terms and conditions of participation were presented during the LEx online registration process and accepted by subjects upon enrollment. In order to register, people had to be at least 18 years old and agreed that the evidence that was gathered during experiments would be used for scientific research. They are informed that participation is totally voluntary and that there is a financial compensation which varies between studies. At the LEx, participants are free to decide whether to accept an invitation and participate in a particular study. When participants begin taking part in a study including laboratory sessions, they are mostly unaware of the type of process that awaits them in the lab. We have carefully considered the ethics of conducting the study as described above: In order to be able to pursue the research as intended, it was necessary to achieve a situation in which the participants did not know about the existence of control and treatment groups as well as the actual processes in the laboratory. However, after completing the last part of the study, all members of the treatment groups had been informed about the actual study process and given the opportunity to withdraw their permission for data usage. Participants were also able to withdraw from the study at all times. Contact information of contact persons was made available at each step. The design of the data collection processes, both during the online surveys and in the laboratory, ensured that the information provided could not be assigned to the participating persons at any time. In order to ensure responsible handling of study subjects, a study draft was submitted to the Ethical Committee at Medical Faculty of Leipzig University. The Ethics Committee reviewed the experimental design under ethical, medical-scientific and legal aspects and confirmed that the design as well as the mode of operation comply with the legal regulations and relevant ICH-GCP recommendations for risk-benefit assessment of scientific investigations in humans. The full statement of the Ethics Committee was provided upon submission.

## Results

### Descriptive and inferential statistics

In order to control for potential random confounding that might have occurred despite randomization, we measured various sensible covariates such as age, student status, sex and diet. As we intended to take potential distorting effects of participant personality into account, we furthermore included the Big Five Inventory Short Scale (BFI-10 [29]). It is established that

personality traits as measured by the Big Five Inventory have a great influence on attitudes and behavior [30, 31]. With respect to the nature of our study, we wanted to make sure that personal characteristics do not cause any unobserved systematic differences on the effect of social feedback on private opinions. This approach seems to be warranted as research has already shown that Big Five personality traits have an influence on feedback seeking behavior [32] and that characteristics such as extraversion and agreeableness have an influence on affective polarization towards an opposing opinion group [33]. An overview of all variables can be found in S1 Appendix. The sample analyzed includes all subjects for whom the data collected at the different time points could be merged. In addition, only subjects with valid values for all of the aforementioned variables are taken into account. The resulting sample includes 229 participants.

Table 1 shows that no striking initial differences can be found between members of the control ($n = 57$) and the treatment group ($n = 172$), hereby analyzed as a whole. Moreover, participants in the control condition ($M = 45.8$, $SD = 31.8$) and the treatment condition ($M = 45.0$, $SD = 32.6$) reported, on average, similar private opinions towards the target item statement. These findings are consistently supported by both the corresponding two-sample tests of proportions as well as the two-sample $t$-tests. None of the test results indicate statistically significant differences between both the main groups of participants at the outset of the study.

The histograms and kernel density estimations presented in Fig 2, however, allow for a more detailed look at the initial opinion distributions, while also providing a first impression of the general outcome of the social feedback process that took place in the laboratory: Both groups not only show almost identical mean values at $t_1$, but very similar median values as well. Thus, around half of the control group members reported a rather disagreeing private opinion ($Mdn = 49$, $IQR = 71–15$), the other half positioned themselves on the part of the scale that represents agreement. While the same is true for the treatment group members ($Mdn = 48$, $IQR = 71–14.5$), small differences in the opinion distributions are discernible as participants in the control condition were slightly more likely to take an either neutral stance or lean towards complete agreement; similarity of initial opinion distributions is nonetheless supported by the curves of the kernel density estimations.

**Table 1. Descriptive statistics (control vs. treatment group, $t_1$).**

| Variables | Control Group $M(SD)$ | Treatment Group $M(SD)$ |
|---|---|---|
| **Target Item** (0–100) | | |
| "The killing and eating of [. . .]." | 45.8(31.8) | 45.0(32.6) |
| **Big Five Inventory** (1–5) | | |
| Agreeableness | 3.2(.8) | 3.1(.8) |
| Conscientiousness | 3.6(.8) | 3.6(.8) |
| Extraversion | 3.1(1.0) | 3.3(.9) |
| Neuroticism | 3.1(.9) | 3.0(.9) |
| Openness | 4.0(.8) | 3.9(.9) |
| **Covariates** | | |
| Age in years | 26.4(6.6) | 27.4(7.2) |
| Vegetarians/Vegans | 24.6% | 20.9% |
| Students | 80.7% | 73.8% |
| Males | 33.3% | 31.4% |
| | $n = 57$ | $n = 172$ |

SOURCE: ODSF2018 (https://doi.org/10.6084/m9.figshare.16595378.v1), own calculations.

## Histograms

## Kernel Density Estimations

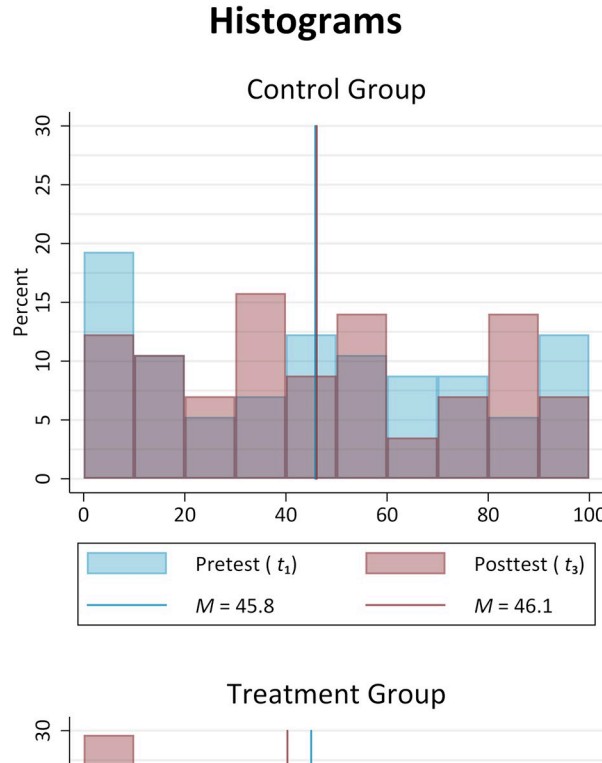

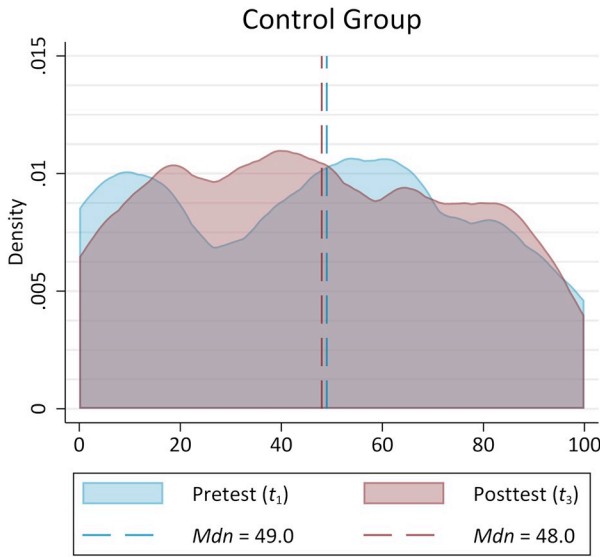

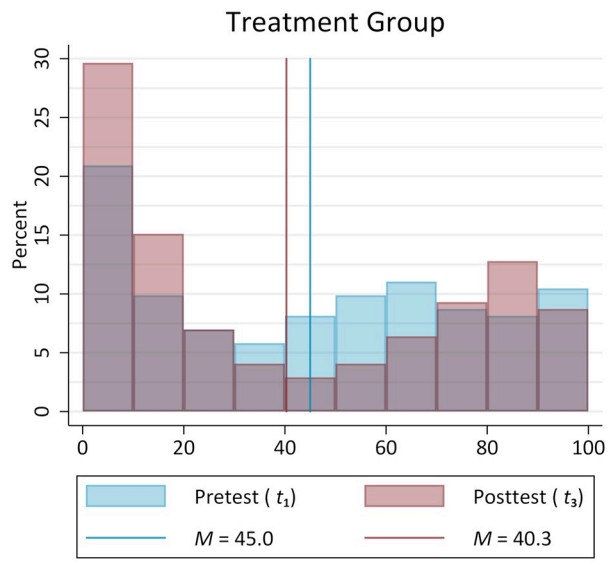

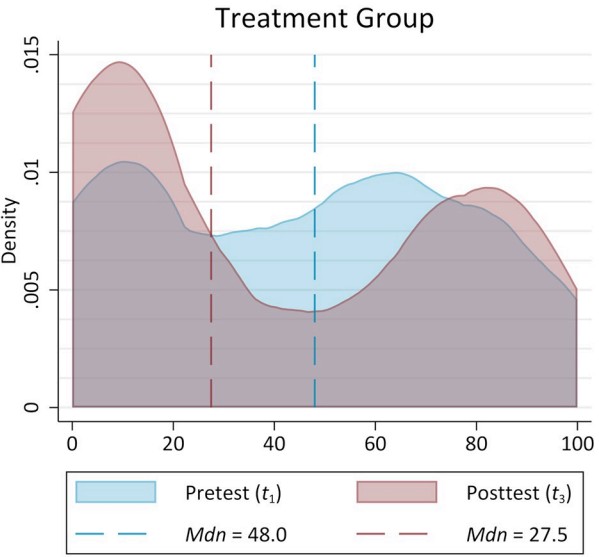

## "The killing and eating of animals is part of human nature."

**Fig 2. Target item distributions (control vs. treatment group).** SOURCE: ODSF2018 (https://doi.org/10.6084/m9.figshare.16595378.v1), own calculations.

Fig 2 moreover illustrates that in the control group the distribution of our target item variable changed slightly, but insubstantially over the course of the study. For members of the treatment group, however, the opinion distribution was subject to a more pronounced change, which is in line with expectations. Conducting a paired-samples $t$-test to compare the opinions reported before ($M = 45.0$, $SD = 32.6$) and after ($M = 40.3$, $SD = 35.4$) the treatment reveals a small yet statistically significant reduction in average agreement, $t(171) = -2.49$, $p = .014$, $d_z =$

**Table 2. Mean values (control vs. treatment groups, $t_1$ vs. $t_3$).**

| | n | $t_1$ M(SD) | $t_3$ M(SD) | Δ M(SD) | 95% CI | Cohen's $d_z$ |
|---|---|---|---|---|---|---|
| Control Group | 57 | 45.8(31.8) | 46.1(29.6) | .3(22.9) | [−5.8, 6.4] | .01 |
| Treatment Group | 172 | 45.0(32.6) | 40.3(35.4) | −4.7(24.7) | [−8.4, −1.0] | −.19 |
| *Positive Feedback* | 37 | 49.2(33.4) | 46.4(35.5) | −2.9(28.3) | [−12.3, 6.6] | −.10 |
| *Negative Feedback* | 54 | 53.9(32.0) | 50.3(35.7) | −3.6(24.0) | [−10.2, 3.0] | −.15 |
| *Mixed Feedback* | 81 | 37.0(31.0) | 30.8(33.1) | −6.3(23.6) | [−11.5, −1.0] | −.26 |

SOURCE: ODSF2018 (https://doi.org/10.6084/m9.figshare.16595378.v1), own calculations.

−.19. A Wilcoxon signed-ranks test furthermore confirms the observable shift in the median value, from $Mdn = 48.0$ ($IQR = 71−14.5$) to $Mdn = 27.5$ ($IQR = 77−7$), to be statistically significant as well, $Z = −2.28$, $p = .023$, $r = −.12$. In fact, the majority of the treated reported disagreeing positions after participating in a laboratory session. The posttest histograms as well as the kernel density estimations also illustrate that while members of the control group were more prone to report less extreme positions at $t_3$, there is a noticeable reduction of neutral opinions and a strengthening of the extreme ends of the scale for members of the treatment group after social feedback was applied, a tendency which has led to a slightly bi-polarized opinion distribution.

Assessing the actual impact of the various social feedback conditions on the private opinion of interest requires a closer look at the differences in intra-group changes, as provided in Table 2 and Fig 3: The reduction in average agreement, from $M = 49.2$ ($SD = 33.4$) at $t_1$ to $M = 46.4$ ($SD = 35.5$) at $t_3$, is smallest ($d_z = −.10$) and insignificant for those who received exclusively positive social feedback during a laboratory session. The same is true for the nonetheless notable shift in the group-specific median value (from $Mdn = 57.0$, $IQR = 73−23$ to $Mdn = 42.0$, $IQR = 77−15$). With a decrease from $M = 53.9$ ($SD = 32.0$) to $M = 50.3$ ($SD = 35.7$) the change of opinion is only somewhat more pronounced in the negative feedback condition ($d_z = −.15$). A comparison of the opinion distributions, as well as their differences after the respective social feedback was applied, in fact reveals surprising similarities between these two treatment groups. In addition to the similar shapes of the initial curves, comparably formed distributions of the private opinion are also to be found in the posttest data. Contrary to the theoretical expectations presented above, however, the impact of the laboratory treatment is strongest for participants in the mixed feedback condition. In this group the reduction in average agreement, from $M = 37.0$ ($SD = 31.0$) to $M = 30.8$ ($SD = 33.1$), is found to reach statistical significance, $t(80) = −2.38$, $p = .020$. Although the effect size is still rather small ($d_z = −.26$), the simultaneous exposure to the two opposing types of social feedback, i.e. positive and negative, has led many of the group members to express strong or even complete disagreement. This is also reflected in the median value, which decreased from $Mdn = 31.0$ ($IQR = 66−8$) to $Mdn = 14.0$ ($IQR = 61−2$) and thus supports the impression of the mixed feedback condition to be the most influential of the three, $Z = −2.138$, $p = .033$, $r = −.17$. Overall, it is striking that across all treatment groups an erosion of neutral positions took place. The application of a social feedback treatment in the laboratory therefore generally led more participants to express more pronounced positions at $t_3$.

We thus observe that although the analyses of group-specific means point to only minor changes in the average opinions of the participants, a closer look at the actual opinion distributions reveals quite broad transitions at the individual level. This becomes even more obvious when further measures and representations are taken into account: The high standard

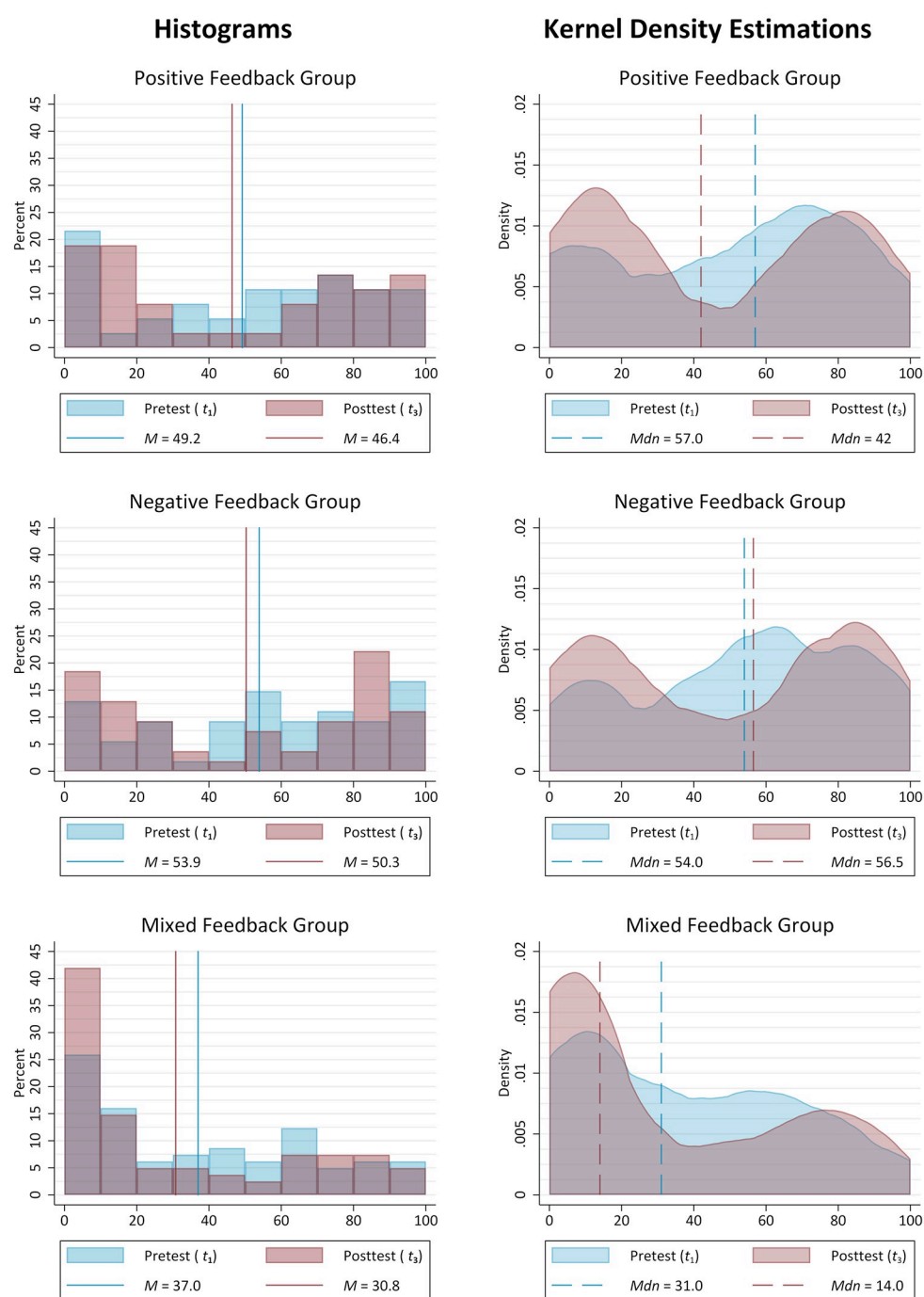

"The killing and eating of animals is part of human nature."

**Fig 3. Target item distributions (treatment groups).** Source: ODSF2018, own calculations.

deviations of the respective mean value differences in Table 2 as well as the box plots, especially of the differences, as presented in Fig 4, illustrate the extent to which participants' individual opinions varied over the course of the study. Ultimately, since the means of the group-specific absolute differences, ranging from $M = 15.1$ ($SD = 19.2$) to $M = 18.2$ ($SD = 21.6$), are

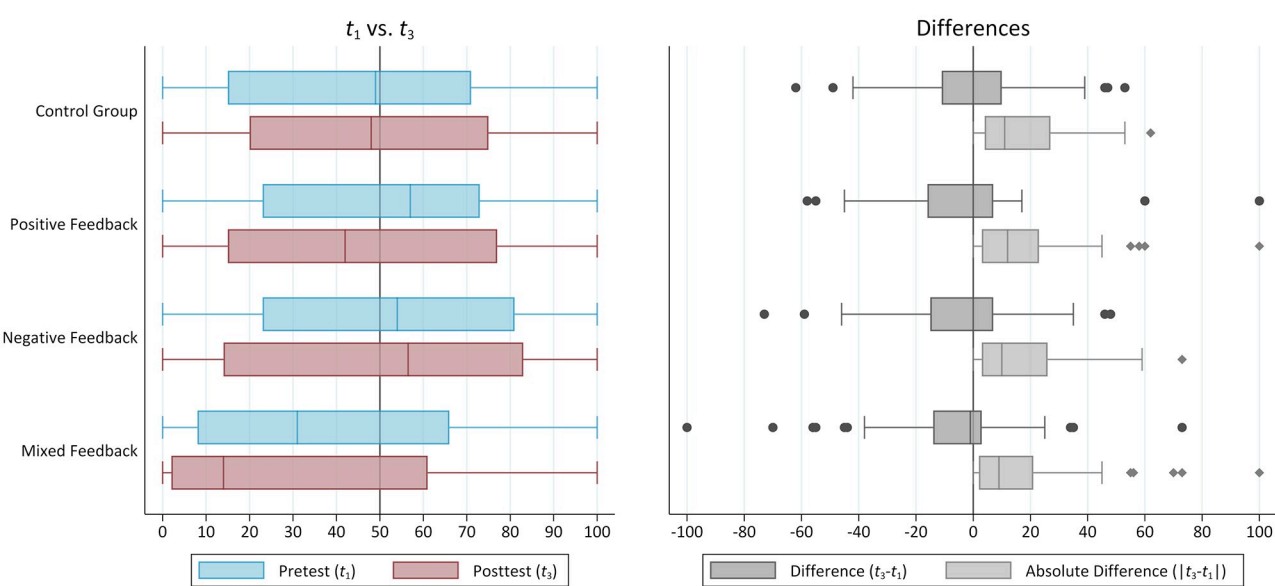

"The killing and eating of animals is part of human nature."

**Fig 4. Boxplots of target item distributions and differences.** SOURCE: ODSF2018 (https://doi.org/10.6084/m9.figshare.16595378.v1), own calculations.

considerably large and quite similar across the four groups under study, the general extent of opinion changes on the individual level seems far from negligible. Therefore, we conclude that there was in fact a quite pronounced opinion volatility in all conditions.

Moreover, a comparison of the various treatment groups with respect to the initial levels of both the target item variable and the covariates indicates that, despite extensive randomization and the seemingly sufficient equality of the control and treatment group members in general (Table 1), the compositions of the three treatment groups turn out to be different in some respects (see Table 3). While 40.7% of the negative feedback group members identified

**Table 3. Descriptive statistics (control vs. treatment groups, $t_1$).**

| Variables | Control $M(SD)$ | Positive $M(SD)$ | Negative $M(SD)$ | Mixed $M(SD)$ |
|---|---|---|---|---|
| **Target Item** (0–100) | | | | |
| "The killing and eating of [. . .]." | 45.8(31.8) | 49.2(33.4) | 53.9(32.0) | 37.0(31.0) |
| **Big Five Inventory** (1–5) | | | | |
| Agreeableness | 3.2(.8) | 2.9(.8) | 3.0(.8) | 3.2(.7) |
| Conscientiousness | 3.6(.8) | 3.8(.8) | 3.6(.8) | 3.5(.8) |
| Extraversion | 3.1(1.0) | 3.5(.9) | 3.3(.9) | 3.2(.9) |
| Neuroticism | 3.1(.9) | 2.9(1.1) | 2.9(1.0) | 3.2(.8) |
| Openness | 4.0(.8) | 3.9(1.0) | 3.8(1.0) | 3.8(.9) |
| **Covariates** | | | | |
| Age | 26.4(6.6) | 26.1(6.3) | 29.3(9.4) | 26.8(5.7) |
| Vegetarians/Vegans | 24.6% | 18.9% | 14.8% | 25.9% |
| Students | 80.7% | 75.7% | 63.0% | 80.2% |
| Males | 33.3% | 29.7% | 40.7% | 25.9% |
| | $n = 57$ | $n = 37$ | $n = 54$ | $n = 81$ |

SOURCE: ODSF2018 (https://doi.org/10.6084/m9.figshare.16595378.v1), own calculations.

themselves as male, this is true for the smaller share of 25.9% of the participants in the mixed feedback condition. This difference, however, is not statistically significant at the 5% level. On the other hand, 80.2% of the latter were students at the time of study participation, in contrast to only 63.0% of the subjects in the negative feedback group, $z = -2.22$, $p =.026$, that also shows to be statistically significantly different from the control group (80.7%), $z = 2.08$, $p =.037$. The members of the mixed feedback group agreed significantly less with the target item statement at the beginning of the study than did the individuals in the negative feedback group, $t(113) = 3.03$, $p =.003$. In addition, a statistically significant difference between the control and the positive feedback group was found for the personality trait of extraversion, $t(82) = -2.02$, $p =.047$. This suggests that the second randomization step did not lead to almost identical groups.

## Linear mixed-effects regression

As a result of the pretest-posttest-design, the opinions of each participant collected at different times are represented by two data points per item. Assuming that unobserved subject-specific characteristics existed that remained constant throughout the study period, these values cannot be considered as independent observations. Moreover, actual differences in group compositions, as identified in the previous section, argue against causal inferences solely based on direct comparisons of the various groups. And in addition, participants' self-assignment to the various laboratory sessions made it likely that subjects who chose to participate in a specific session, as for instance on a Saturday morning, also might have shared unknown and unobserved characteristics. Ultimately, hypotheses **H1** and **H2** imply that the direction of a potential social feedback treatment effect depends on the initial position of the person whose opinion is subject to judgment. It is therefore appropriate to separately analyze the two fundamental baseline conditions in which either a disagreeing private opinion was held at $t_1$ or an agreeing position existed.

In order to account for clustering on the session-level, repeated measurements on the subject-level, differences in group composition, and the directional implications of the hypotheses we estimated three multiple three-level linear mixed-effects regression models on the subjects' opinion regarding the target item statement. While model A incorporates those participants who submitted a disagreeing private opinion ($y_{i_{t_1}} < 50$, $n = 120$) towards the target item statement at the beginning of the study, model B consists of participants which reported an agreeing initial position ($y_{i_{t_1}} \geq 50$, $n = 109$). Both the models A and B are of particular relevance for the evaluation of our hypotheses and provided in full in Table 5 alongside a third model C, which incorporates the entire sample ($n = 229$).

In each of the models the laboratory sessions represent the top level (level 3 groups). Repeated measurements of the target opinion on the lowest level (level 1 observations) are considered to be nested within subjects (level 2 groups). Predictor variables and covariates are included as fixed effects, with the former being binary variables indicating the specific experimental group membership. As a consequence, the associated partial regression coefficients can be interpreted in contrast to the respective groups' state at $t_1$. The Big Five personality traits, vegetarian or vegan diet, student status, sex, and age are included as covariates; BFI-10 variables and age are z-transformed. After converting the data from wide to long format, the dependent variable is composed of the individuals' target item measurements at times $t_1$ and $t_3$. For each participant it thus contains two integer values ranging from 0 (complete disagreement) to 100 (complete agreement), the first one resulting from the pretest at $t_1$ and the second measured during the posttest at $t_3$. Table 4 illustrates the structure of the dataset used for the multilevel analysis presented within this section. Unobserved heterogeneity on the session-level and unobserved differences between subjects are controlled by specifying respective

**Table 4. Illustration of the dataset (long format).**

| ID | Time of Observation | Dependent Variable | Control Group | Positive Feedback | Negative Feedback | Mixed Feedback | Age (z-Score) | ... |
|----|---------------------|--------------------|---------------|-------------------|-------------------|----------------|---------------|-----|
| 1 | 0 | 50 | 0 | 0 | 0 | 0 | -.58 | ... |
| 1 | 1 | 47 | 1 | 0 | 0 | 0 | -.58 | ... |
| 2 | 0 | 40 | 0 | 0 | 0 | 0 | -.44 | ... |
| 2 | 1 | 100 | 0 | 1 | 0 | 0 | -.44 | ... |
| 3 | 0 | 27 | 0 | 0 | 0 | 0 | -.72 | ... |
| 3 | 1 | 22 | 0 | 0 | 0 | 1 | -.72 | ... |
| ... | ... | ... | ... | ... | ... | ... | ... | ... |

random intercepts. In addition to that, each participant is assumed to have a subject-specific random slope, hence allowing for varying treatment effects among subjects. In general, the corresponding likelihood-ratio tests show that introducing random intercepts on the session- and subject-level led to models that are statistically significantly superior to their linear equivalents, which are blind to the specifics of our multilevel data structure.

**Fixed effects.** A first look at the fixed-effects part of Table 5 shows that both models have differing overall-intercepts, which naturally results from separating the two specific sample cutouts of interest. For both the initially disagreeing participants (estimate = 21.06, $p <$.001, 95% CI [13.56, 28.55]) and the sample of individuals formerly found on agreeing positions (estimate = 73.88, $p <$.001, 95% CI [66.34, 81.41]), intercepts are estimated to be close to the middle of their respective part of the scale.

There is a fairly clear movement of private opinion towards the neutral middle of the full target item scale for the untreated control group in both models. As can be seen in Table 5 and Fig 5, the agreement with the target item statement increased, on average, by 7.51 ($p =$.056, 95% CI [−.20, 15.22]), when there was a disagreeing private opinion at the beginning and no social feedback was applied, whereas in model B the agreement decreased to a similar extent (estimate = −7.29, $p =$.047, 95% CI [−14.50, −.08]). In general, it can be observed that although the tendency towards the neutral positions is evident in almost all of the treatment groups, it's extent varies depending on the respective feedback condition. Moreover, social feedback generally resulted in formerly disagreeing subjects tending somewhat less towards the center of the original target item scale than is the case for the treated in model B.

For both, positive (estimate = 5.69, $p =$.465, 95% CI [−9.56, 20.945]) and negative social feedback (estimate = 3.88, $p =$.386, 95% CI [−4.89, 12.65]), the tendency towards the center is slightly reduced in model A. With an estimated average reduction in agreement of −1.08 ($p =$.638, 95% CI [−5.59, 3.42]), the comparably strongest influence on the change of opinion once again can be found for the mixed feedback condition. In the sample of those who gave an agreeing private opinion at $t_1$, a striving towards the neutral part of the target item scale is also evident. However, while in model A social feedback slows down the movement towards the scale center and binds the participants more strongly to the disagreeing areas, in model B the application of social feedback has had a slightly catalyzing effect that drives the movement towards the less agreeing parts of the scale. The different treatment groups thus tend to show slightly more pronounced decreases, with the most prominent effect again in the mixed feedback group (estimate = −16.14, $p =$.005, 95% CI [−27.33, −4.96]). When comparing the confidence intervals of the group-specific changes in private opinion over time (Fig 5), though, despite the observable trends neither model A nor B show statistically significant differences between the various groups.

**Random effects.** For individuals in the initially disagreeing subsample, the standard deviation of the random session-intercept ($SD$ = 4.13, 95% CI [1.18, 14.51]) indicates statistically

**Table 5. Results of the multiple three-level linear mixed-effects regressions on the subjects' opinion towards the target item statement.**

| | Model A ($y_{i_{t_1}} < 50$) | Model B ($y_{i_{t_1}} \geq 50$) | Model C (full sample) |
|---|---|---|---|
| **Random Effects** | **SD** (*SE*) | **SD** (*SE*) | **SD** (*SE*) |
| Session Intercept | 4.13***(2.65) | .00(.) | .00(.) |
| Subject Intercept | 13.21***(1.62) | 13.95***(1.68) | 24.67***(1.50) |
| Social Feedback Treatment | | | |
| Subject: Control Group | 17.40***(4.06) | 16.03***(3.67) | .00(.00) |
| Subject: Positive Feedback | 28.39***(6.32) | 20.72***(4.30) | 13.77***(6.04) |
| Subject: Negative Feedback | 16.01***(4.14) | 23.96 ***(3.72) | 9.24***(6.35) |
| Subject: Mixed Feedback | 10.08***(3.45) | 29.14***(1.68) | 4.15***(11.53) |
| **Fixed Effects** | **Estimate** (*SE*) | **Estimate** (*SE*) | **Estimate** (*SE*) |
| Intercept | 21.06***(3.82) | 73.88***(3.84) | 49.55***(4.58) |
| Social Feedback Treatment | | | |
| Control Group | 7.51+(3.93) | −7.29*(3.68) | 1.14(2.89) |
| Positive Feedback | 5.69(7.78) | −10.01*(4.99) | −1.96(4.22) |
| Negative Feedback | 3.88(4.47) | −8.21+(4.50) | −2.17(3.23) |
| Mixed Feedback | −1.08(2.30) | −16.14**(5.71) | −8.22**(2.49) |
| Big Five Inventory | | | |
| Agreeableness† | −1.26(1.53) | −2.48(1.58) | −5.33**(1.87) |
| Conscientiousness† | −2.27(1.43) | −.45(1.68) | −1.91(1.86) |
| Extraversion† | 1.15(1.72) | 1.73(1.59) | 0.05(2.00) |
| Neuroticism† | −.53(1.67) | −.89(1.66) | −1.19(2.03) |
| Openness† | −.73(1.75) | −1.50(1.47) | −3.21+(1.94) |
| Diet | | | |
| *Neither Vegetarian nor Vegan* | | | |
| Vegetarian or Vegan | −10.60**(3.19) | −12.29*(5.38) | −27.98***(4.49) |
| Student Status | | | |
| *Not a Student* | | | |
| Student | .32(4.18) | .46(4.44) | .88(5.24) |
| Sex | | | |
| *Female* | | | |
| Male | 1.78(3.54) | 2.27(3.46) | 3.33(4.27) |
| Age† | .13(2.11) | .88(1.69) | 2.35(2.24) |
| AIC | 2048.9 | 1888.8 | 4251.0 |
| $\chi^2$ (LR Test vs. Linear Model) | 48.09 | 56.52 | 133.75 |
| *n* | 120 | 109 | 229 |

Source: ODSF2018 (https://doi.org/10.6084/m9.figshare.16595378.v1), own calculations. Notes: *N* observations = 458, *N* subjects = 229, *N* sessions = 19, standard errors in parentheses, † z-transformed, reference categories in italics, + $p \leq .10$, * $p \leq .05$, ** $p \leq .01$, *** $p \leq .001$ (two-tailed). The restricted maximum likelihood estimations (REML) were performed using the `mixed` command as provided in Stata 15.1.

significant variations between baseline levels in the various laboratory sessions. This cannot be found for the originally agreeing subjects. However, random intercepts show quite strong and statistically significant variability at the subject-level in model A (*SD* = 13.21, 95% CI [10.38, 16.80]) as well as in model B (*SD* = 13.95, 95% CI [11.01, 17.67]). In all groups a strong between-subjects variation of the slopes is apparent. Thereby, its extent is similar between control group members in both models, with *SD* = 17.40 (95% CI [11.01, 27.50]) in model A and *SD* = 16.03 (95% CI [10.24, 25.10]) in model B. In contrast, a different picture emerges for the treatment groups: Whereas in model A the between-subjects variability of the slopes is highest

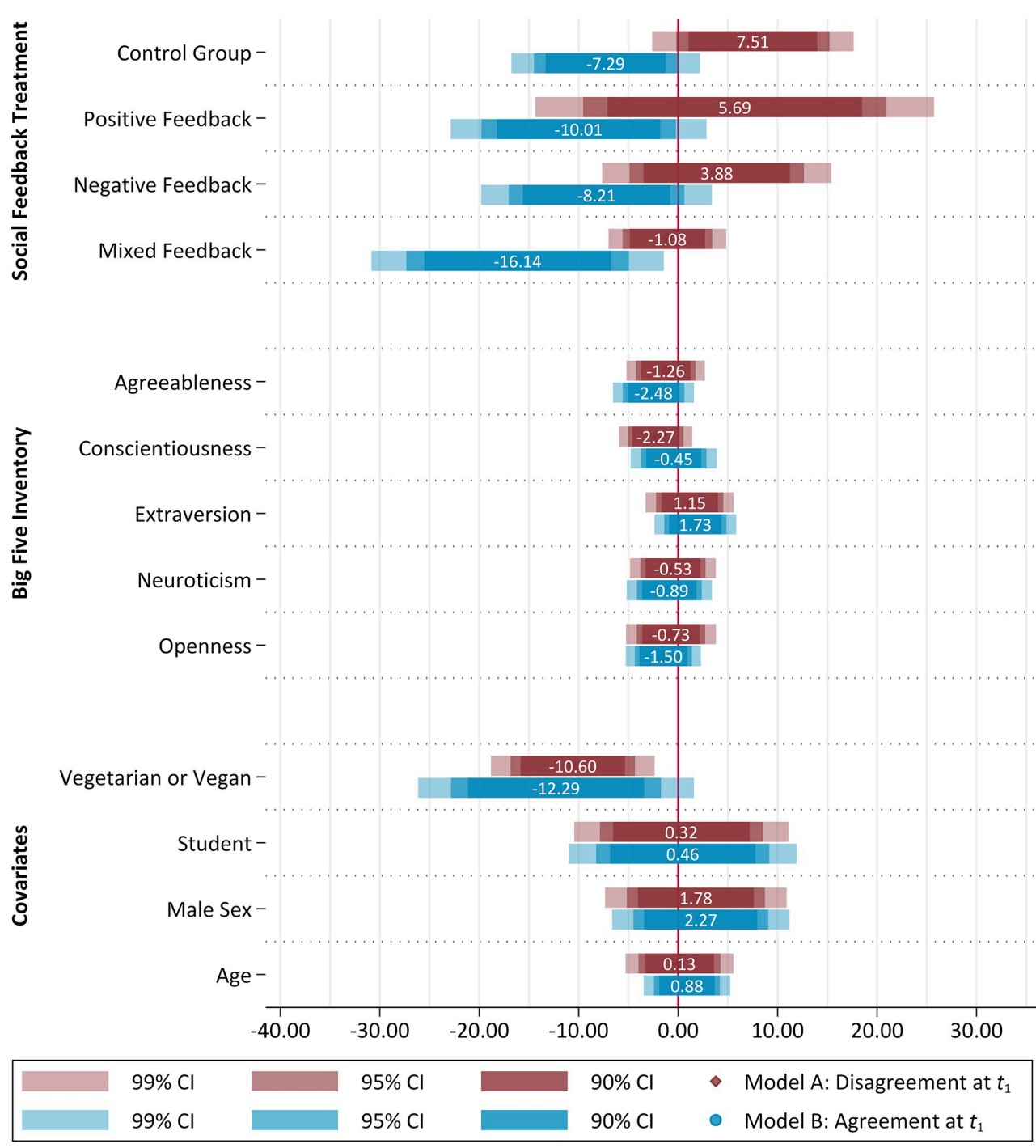

**Fig 5. Coefficients plot of the multiple three-level linear mixed-effects regression.** SOURCE: ODSF2018 (https://doi.org/10.6084/m9.figshare. 16595378.v1), own calculations. NOTES: *N* observations = 458, *N* subjects = 229, *N* sessions = 19. BFI-10 indices and age are included as z-transformed variables, intercepts are omitted. The graph was generated using the `coefplot` package for Stata [34].

in the positive feedback group (*SD* = 28.39, 95% CI [18.35, 43.92]) and lowest in the mixed feedback group (*SD* = 10.08, 95% CI [5.16, 19.71]), things turn out the other way around in model B, in which we estimate *SD* = 20.72 (95% CI [13.79, 31.13]) for the former and *SD* = 29.14 (95% CI [21.57, 39.37]) for the latter.

## Discussion

The question of how and to what extent people influence each other in social exchange processes has been of interest for a long time. It is of growing relevance as large segments of people nowadays are involved in online communication and interaction settings in which they find themselves exposed to social judgments by unknown others.

While the debate about potential implications of these novel exchange processes is carried out with fervor, empirical evidence is scarce. This paper is contributing to this discussion with an experimental study that analyzed the influence of social feedback on actual private opinions. Our research questions and hypotheses were inspired by social feedback theory [18] which centers around George Homans' idea that human beings adapt their behavior according to the reactions it evokes from others [3].

We derived and empirically tested three hypotheses. **H1** asserts that social feedback perceived as positive causes recipients to push their private opinion further in the direction of the original position, whereas **H2** states that people who receive negative social feedback will strive towards the opposite direction of their initial opinion. We furthermore expected (**H3**) that the administration of both positive and negative feedback would result in no change in the private opinion of interest. The hypotheses were tested through an experimental design that combined a preceding and a follow-up online survey with a laboratory session in which participants were asked to evaluate a statement regarding meat consumption and subsequently received a social feedback stimulus of either positive, negative or mixed content.

The descriptive analyses show that both the control and the treatment group in general shared very similar target opinion distributions at the outset; with each half of the participants reporting either rather agreeing and disagreeing opinions. Following the social feedback treatment, there was only a small negative effect on the average agreement with the target item statement in both the positive and the negative feedback group. Contrary to our expectation, however, a statistically significant reduction in agreement occurred among subjects that got randomly assigned to the mixed feedback treatment. Besides, it is striking that we observe an erosion of neutral positions in all three social feedback conditions which leads to opinion distributions that are notably bi-polarized.

In order to test our hypotheses according to the requirements of our data, we estimated several multiple three-level linear mixed-effects regression models. For the control group it becomes apparent that participants who originally disagreed with the target item as well as those who initially agreed moved to an almost equal extent towards the respective opposite side of the scale. In principle, this also applies to participants who received either unambiguous positive or negative social feedback. If an equal mixture of positive and negative feedback statements is applied, however, it is predicted that the natural striving to the middle of the scale will be affected: While initially disagreeing individuals are predicted to hold on to their disagreeing stance, it is estimated that individuals that agreed in the beginning move even further towards the less agreeing domains than subjects in other feedback conditions.

In principle, both research questions, namely whether social feedback is influential and whether this influence depends on the type of feedback, can be answered affirmatively. However, the specific effects as claimed in the hypotheses could not be confirmed. Our findings are nonetheless remarkable exactly because they contradict the theoretical expectations that are plausible and obvious at first glance. This is particularly evident in light of the fact that it was not merely spontaneous public adjustments that had been observed, but actual and persistent changes in private opinions measured several days to weeks after the social feedback was applied. It is all the more astonishing as participants were led to believe that they had received this feedback within the context of a singular interaction with unknown and unidentifiable

strangers in a sanction free lab environment. These unexpected initial results motivate further investigations as well as replication studies. This is also demanded by the marked degree of opinion volatility of which only a small proportion can be attributed to the social feedback alone. Since private opinions can vary for manifold external and internal reasons, this may naturally result from the repeated measurement design spanning over several weeks. Overall, the participants' opinions do not seem to have been especially stable and persistent, but have undergone variations based on presumably many additional unobserved factors.

For future studies, we suggest conceptual replications and the further development of the presented experiment in order to validate our findings. First, with regards to content, it would be important to extend the range of opinion statements that participants receive feedback for. In order to get a first and fundamental insight into the effect of social feedback on private opinions, we chose the alleged study object with caution, aiming for a topic that was neither strongly polarized nor unfamiliar to participants. We did not however measure the relative importance that the attitude towards the killing and eating of animals holds for participants which is one of the limitations of our study. With regards to the most urgent questions of opinion polarization and its potentially detrimental effects on social cohesion it would be important to test the effect of social feedback on convictions that are of crucial importance to participants' identities and therefore emotionally charged. In line with the theoretical reasoning that we presented above we expect that even in this case exposure to social feedback will generally result in a change in opinion in the direction that participants believe to be rewarded by their social surroundings. Yet for this change to be distinct and lasting we expect that the social feedback stimulus would have to be implemented more frequently and by a larger number of feedback donors. It is also unclear whether the expected assimilation of these more relevant opinions would occur in a situation that is completely anonymous and sanction free. An experimental set-up which varies the degree of anonymity and potential punishment (as for example the prospect of face to face interaction in the future would provide) could shed some light on these questions which we hope will be addressed by future research. Secondly, and on a methodical note, we have to acknowledge that the slider used in the online questionnaires itself may have been a source of dispersion. In retrospect, it seems to be quite difficult to give the exact same opinion twice using such a sensitive response format that incorporates around 100 possible opinion values, even if the opinion has not actually changed. In this respect, some noise could have been introduced by the sliders alone and may already be eliminated by implementing a different response format.

We hope that our research contributes to the necessary debate on the influence of social feedback on opinion changes. As networks of information and opinion exchange are likely to gain even more density and importance in the years to come and as empirical evidence on the influence of social feedback on opinions is scarce, we believe that this field of research deserves more attention than it has received so far.

## Supporting information

**S1 Table. Social feedback statements.** List of all social feedback statements used during the laboratory treatment process.
(PDF)

**S1 Appendix. Variable report.** Explanations of all variables in the dataset.
(PDF)

## Author Contributions

**Conceptualization:** Marcel Sarközi, Sven Banisch.

**Data curation:** Marcel Sarközi.

**Formal analysis:** Marcel Sarközi.

**Investigation:** Marcel Sarközi.

**Methodology:** Marcel Sarközi.

**Project administration:** Marcel Sarközi, Sven Banisch, Roger Berger.

**Resources:** Stephanie Jütersonke.

**Supervision:** Sven Banisch, Stephan Poppe, Roger Berger.

**Visualization:** Marcel Sarközi.

**Writing – original draft:** Marcel Sarközi, Stephanie Jütersonke.

**Writing – review & editing:** Marcel Sarközi, Stephanie Jütersonke, Sven Banisch, Stephan Poppe, Roger Berger.

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
