## [Decision Letter · Decision Letter 0]

6 Jun 2022

PONE-D-21-30777The effects of social feedback on private opinions. Empirical evidence from the laboratory.PLOS ONE

Dear Dr. Sarközi,

Thank you for submitting your manuscript to PLOS ONE. After careful consideration, we feel that it has merit but does not fully meet PLOS ONE’s publication criteria as it currently stands. Therefore, we invite you to submit a revised version of the manuscript that addresses the points raised during the review process.

This is an interesting topic and your finding of polarization of initial opinion is especially relevant in today’s environment. My major concern is your analysis, which should focus on the change/difference instead of the level.

In your linear models, you stated that your dependent variable is composed of the individuals’ target item measurements at times t1 and t3. Please clarify the specific measurement of your DV. Is it the difference between t1 and t3, the average, or the sum? Or is it the opinion at t3?In Table 2 below, could you explain why at t1, the mixed feedback group is significantly lower than other groups? This will not change your results if the analysis focuses on the difference between t1 and t3. In addition, in this group, you have a much higher number of participants.You provided means of the group-specific absolute differences (ranging from M = 15.1 (SD = 19.2) to M = 18.2 (SD = 21.6), are considerably large and quite similar across the four groups). Please explain the significance of this measurement. I assume it is the absolute difference at the individual level for each subject. I think this measurement is superior to the regular difference if you follow up with more analysis. For example, you can break it down in 3 groups, 1) more positive opinion after t2,  t3-t1>0, 2) more negative, t3-t1<0, 3) no change, t3-t1=0. This will not only show the direction but also the magnitude of the change.You find the target item variable and the covariates significantly different despite “seemingly sufficient equality of the control and treatment group members”. You should include analysis to make sure whether the means are statistically different. In your Linear Mixed-Effects Regression, also think about including these covariates as control variables.  Please submit your revised manuscript by Jul 21 2022 11:59PM. If you will need more time than this to complete your revisions, please reply to this message or contact the journal office at plosone@plos.org. Please include the following items when submitting your revised manuscript:A rebuttal letter that responds to each point raised by the academic editor and reviewer(s). You should upload this letter as a separate file labeled 'Response to Reviewers'.A marked-up copy of your manuscript that highlights changes made to the original version. You should upload this as a separate file labeled 'Revised Manuscript with Track Changes'.An unmarked version of your revised paper without tracked changes. You should upload this as a separate file labeled 'Manuscript'.

We look forward to receiving your revised manuscript.

Kind regards,

Ning Du

Academic Editor

PLOS ONE

Journal Requirements:

2. Please provide additional details regarding participant consent. In the Methods section, please ensure that you have specified (1) whether consent was informed and (2) what type you obtained (for instance, written or verbal). If your study included minors, state whether you obtained consent from parents or guardians. If the need for consent was waived by the ethics committee, please include this information.

Additional Editor Comments (if provided):

This is an interesting topic and your finding of polarization of initial opinion is especially relevant in today’s environment. My major concern is your analysis, which should focus on the change/difference instead of the level.

1) In your linear models, you stated that your dependent variable is composed of the individuals’ target item measurements at times t1 and t3. Please clarify the specific measurement of your DV. Is it the difference between t1 and t3, the average, or the sum? Or is it the opinion at t3?

2) In Table 2 below, could you explain why at t1, the mixed feedback group is significantly lower than other groups? This will not change your results if the analysis focuses on the difference between t1 and t3. In addition, in this group, you have a much higher number of participants.

3) You provided means of the group-specific absolute differences (ranging from M = 15.1 (SD = 19.2) to M = 18.2 (SD = 21.6), are considerably large and quite similar across the four groups). Please explain the significance of this measurement. I assume it is the absolute difference at the individual level for each subject.I think this measurement is superior to the regular difference if you follow up with more analysis. For example, you can break it down in 3 groups, 1) more positive opinion after t2, t3-t1>0, 2) more negative, t3-t1<0, 3) no change, t3-t1=0. This will not only show the direction but also the magnitude of the change.

4) You find the target item variable and the covariates significantly different despite “seemingly sufficient equality of the control and treatment group members”. You should include analysis to make sure whether the means are statistically different. In your Linear Mixed-Effects Regression, also think about including these covariates as control variables.

Reviewers' comments:

Reviewer's Responses to Questions

**Comments to the Author**

1. Is the manuscript technically sound, and do the data support the conclusions?

Reviewer #1: Yes

2. Has the statistical analysis been performed appropriately and rigorously? 

Reviewer #1: Yes

3. Have the authors made all data underlying the findings in their manuscript fully available?

Reviewer #1: Yes

4. Is the manuscript presented in an intelligible fashion and written in standard English?

Reviewer #1: Yes

5. Review Comments to the Author

Reviewer #1: This is an interesting study of the effects (if any) of social feedback on attitude change. The experiment is well-designed and the results are written up clearly, for the most part.

Small thing I would like to see:

More discussion of the scatterplots (Fig 5), especially interpretation of their meaning. I have not seen this particular form of visualization in the context of experiment-to-experiment change, and it was a bit confusing at times.

Some of the language of results could be cleaned up. For example, on Page 13 authors state that "Once again it is the mixed feedback condition 441 in which the most striking effect is seen, with an estimated average reduction in 442 agreement of −1.08." While I parsed apart that what the authors (I think) were referring to is an interesting *non* effect, stating that an effect exists in the presence of a non-effect is confusing. I would rephrase this.

Bigger things:

The authors do well to select an issue that, while contentious (humans eating meat), is not polarized in terms of ideology and partisanship in the European context. Obviously an issue cueing ideology (say, EU membership issues) would change the design quite a lot. However, I would like to see discussion of what the expectation would be in that context, at least in the discussion, given that much of the manuscript is motivated by increasing opinion polarization (supposedly) driven by online communicative domains.

How does motivated reasoning fit into this theoretical framework? If people have strongly-held attitudes on this (or any) issue going into the experiment, that is going to dramatically alter how they perceive positive or negative feedback dramatically, I would think. For example, if I say that humans are meant to eat meat at 90% on the temperature scale, my response to negative feedback might be very different than someone who goes into the lab at 60%, or 40%. Discussion of how dissonance induced by a countervailing attitude is necessary.

Moreover, I think this would be able to be analyzed given the data at hand - I would like the authors to address this by examining not only movement in opinions in the face of pro-/counterattitudinal social cues, but also how those moves (if any) are influenced by how strongly-held those opinions are at the outset. I could imagine some of this being taken care of by presenting plots of random slopes, split (by graph) across participants with strongly- versus weakly-held attitudes.

The Big 5: Why? The personality battery first shows up in the descriptive results (Page 13 of the reviewer copy) but no discussion is made prior to this, or even during the analysis, as to why we would expect the Big Five to be important to control for. More discussion is needed on this point.

6. PLOS authors have the option to publish the peer review history of their article (what does this mean?). If published, this will include your full peer review and any attached files.

Reviewer #1: No

---

## [Author Response · Author response to Decision Letter 0]

5 Sep 2022

(1) In your linear models, you stated that your dependent variable is composed of the individuals’ target item measurements at times t1 and t3. Please clarify the specific measurement of your DV. Is it the difference between t1 and t3, the average, or the sum? Or is it the opinion at t3?

Response: Thanks for pointing out that the description of the dependent variable is not clear. We have made changes accordingly. The analysis focuses on the differences between t1 and t3. As you correctly note, the dependent variable is composed by the repeated target item measurements at time points t1 and t3, i.e. before as well as after the lab sessions. In order to account for the hierarchical structure of the data, the data set has been put into long format, so that instead of having one variable per time point of measurement, a single dependent variable contains the two measurements of opinion for each participant. Thus, we can observe the inter-group differences in the intra-group changes of opinion. In the manuscript, the specifics of the data set structure are now illustrated in Table 4.

(2) In Table 2 below, could you explain why at t1, the mixed feedback group is significantly lower than other groups? This will not change your results if the analysis fo-cuses on the difference between t1 and t3. In addition, in this group, you have a much higher number of participants.

Response: Unfortunately, we cannot explain why the mixed feedback group is composed of participants that on average showed a significantly lower agreement at the beginning of the study. The specific treatment group membership was manifested in the laboratory when participants were randomly assigned their feedback statements (this was the second ran-domization step). A subsequent check of the z-Tree code showed that the random assignment of feedback statements was flawless and indeed purely random. Thus, the difference between the mixed feedback group and the other groups has to be considered a product of coincidence.

The random assignment of social feedback statements initially resulted in four groups: One group of individuals who received exclusively positive feedback, one with exclusively negative feedback, one in which the fictitious group member A’s statement was positive und C’s was negative, and one group in which A appeared to give negative feedback and C positive feedback. Thus, randomization initially created four groups of approximately equal size. 

As a first descriptive analysis showed, the two mixed feedback groups are composed in a very similar way of people who had a lower average agreement with the target item at the beginning of the study – compared to the people in the positive and negative feedback groups; overall, the mixed feedback groups showed to be similar to each other. Furthermore, we have no theoretical assumptions about how the effect of social feedback differs when the order of feedback statements presented simultaneously on the same page is reversed. For these reasons, we decided to group all individuals who received a positive and a negative feedback statement into a single group, the mixed feedback group. This led the latter to be larger in size than the positive and the negative feedback group.

In general, differences in group sizes also arose, for example, because participants did not take part in the last survey or entered their ID codes incorrectly, so that it was not possible to merge their data properly.

(3) You provided means of the group-specific absolute differences (ranging from M = 15.1 (SD = 19.2) to M = 18.2 (SD = 21.6), are considerably large and quite similar across the four groups). Please explain the significance of this measurement. I assume it is the absolute difference at the individual level for each subject. I think this measurement is superior to the regular difference if you follow up with more analysis. For example, you can break it down in 3 groups, 1) more positive opinion after t2, t3-t1>0, 2) more negative, t3-t1<0, 3) no change, t3-t1=0. This will not only show the direction but also the magnitude of the change.

Response: By reporting the average group-specific absolute differences, we want to point out the phenomenon that considering only the average group-specific differences – as provided in Table 2 – can lead to an underestimation of the extent of opinion volatility at the individual level. For example, for the control group (M_t1= 45.8 and M_t3 = 46.1), there is such a small change in average agreement that one could think that the participants’ opinions remained almost unchanged over the course of the study. However, some control group members reduced their agreement, while others increased their agreement to a comparable extent. Thus, the aggregation, i.e. the mean value, creates the impression that the opinions had hardly changed, while in fact there are significant differences at the individual level. 

The average absolute differences provide us with the means to show the extent to which, on average, there was a change of opinion for each person in a group, regardless of its direction. This is to show that, in general, a comparably high opinion volatility can be found in all groups studied; also in the control group, which was not exposed to any social feedback. The difference between the groups, however, can be seen in the fact that the directions of individual changes of opinion had been influenced by social feedback – as in the case of the mixed feedback group, in which on average a reduction in agreement occurred, i.e. increases and decreases in agreement on the individual level did not cancel each other out, but rather the decreases outweighed the increases.

Since our hypotheses not only assert that social feedback leads to changes of opinion, but rather assume specific directions of change depending on the type of social feedback, we do not consider the absolute differences to be a suitable measure for assessing the validity of the hypotheses that are at the core of this study.

(4) You find the target item variable and the covariates significantly different despite “seemingly sufficient equality of the control and treatment group members”. You should include analysis to make sure whether the means are statistically different. In your Linear Mixed-Effects Regression, also think about including these covariates as control variables.

Response: Starting at line 322, we address the question of whether the control group and the treatment group, analyzed as a whole, differed from each other, or whether they actually did not show significantly different measures, as we expected based on the random assignment of our subjects. We found that there were no statistically significant differences between the control and treatment group with respect to the target item, the Big Five Inventory, and the control variables age, diet, student status, and gender (see Table 1). We did not report the insignificant test results in order not to make the text too confusing.

Starting at line 397, we address the phenomenon that despite randomized assignment of social feedback statements, the various treatment groups differ from each other in certain respects. Once again, we thank you for the important comment. In fact, we’ve missed to report the respective test results. Statistically significant differences are now reported in the manuscript. 

All of the aforementioned covariates had been included as control variables in all of the linear mixed-effects models, see Table 5 as well as Fig 5.

(5) More discussion of the scatterplots (Fig 5), especially interpretation of their meaning. I have not seen this particular form of visualization in the context of experiment-to-experiment change, and it was a bit confusing at times.

Response: The discussion of the mean values of the absolute differences and their graphical representation in Fig 4 were intended to emphasize the fact that, overall, a high degree of opinion volatility was found. This was also to be underlined by the scatterplot, which visualizes all actual individual changes of the target opinion over time and reveals that clear patterns of change can hardly be identified. We agree that this is perhaps too much fuss over a less informative fact and have thus decided to remove the scatterplot in order to avoid confusing redundancy.

(6) Some of the language of results could be cleaned up. For example, on Page 13 authors state that “Once again it is the mixed feedback condition 441 in which the most striking effect is seen, with an estimated average reduction in 442 agreement of −1.08.” While I parsed apart that what the authors (I think) were referring to is an interesting *non* effect, stating that an effect exists in the presence of a non-effect is confusing. I would rephrase this.

Response: Thanks for pointing this out, we’ve changed the wording accordingly.

(7) The authors do well to select an issue that, while contentious (humans eating meat), is not polarized in terms of ideology and partisanship in the European context. Obviously an issue cueing ideology (say, EU membership issues) would change the design quite a lot. However, I would like to see discussion of what the expectation would be in that context, at least in the discussion, given that much of the manuscript is motivated by increasing opinion polarization (supposedly) driven by online communicative domains.

Response: Thank you for giving us the opportunity to elaborate on this point. It is without doubt necessary that a validation of our design and findings includes topics that are linked to participants’ identities more closely and therefore more emotionally charged. As our study is (to our knowledge) the first laboratory study which focusses on measuring the direct mechanism that lead people to changing their private opinion after being presented with anonymous social feedback, we wanted to start with a topic that was neither polarized nor unknown to participants and wouldn’t activate other mechanisms (e.g. Backfire effect). We do believe that integrating a topic that is more directly linked to participants´ identities and therefore of greater importance to them would still yield the same effects in the long run, but that the stimulus in this case would have to be more frequent and the number of participant donors would have to be greater. It is a point that we will consider in our future research. In pointing out these assumptions we have included the following paragraph in the discussion of our manuscript. 

(8) 4. How does motivated reasoning fit into this theoretical framework? If people have strongly-held attitudes on this (or any) issue going into the experiment, that is going to dramatically alter how they perceive positive or negative feedback dramatically, I would think. For example, if I say that humans are meant to eat meat at 90% on the temperature scale, my response to negative feedback might be very different than someone who goes into the lab at 60%, or 40%. Discussion of how dissonance induced by a countervailing attitude is necessary.

Response: This comment sparked some discussion for future research in our group as we unfortunately did not include a measure of attitude importance in the present study. 

Assuming that a high value on the attitude scale (indicating agreement or disagreement) equals the importance that the opinion has for the respective person would in our opinion mix up different aspects of opinion that should be treated separately. Just because a person is in complete disagreement with the target statement does not indicate the centrality of the topic. We therefore do not see the opportunity to make testable assumptions about this point on basis of our study design and data base. 

We have updated the manuscript by naming the lack of an appropriate measure of attitude importance as one of the limitations of our study

(9) Moreover, I think this would be able to be analyzed given the data at hand - I would like the authors to address this by examining not only movement in opinions in the face of pro-/counterattitudinal social cues, but also how those moves (if any) are influenced by how strongly-held those opinions are at the outset. I could imagine some of this being taken care of by presenting plots of random slopes, split (by graph) across participants with strongly- versus weakly-held attitudes.

Response: As stated above we do not think that we have the necessary information for making assumptions that include attitude importance. One idea that came to mind in order to make use of what we do have, was to use diet as a proxy of attitude importance, yet our participant groups in that case would be too small to allow for statistically sound testing. This certainly marks a limitation of our study and should be addressed in the future. 

(10) The Big 5: Why? The personality battery first shows up in the descriptive results (Page 13 of the reviewer copy) but no discussion is made prior to this, or even during the analysis, as to why we would expect the Big Five to be important to control for. More discussion is needed on this point.

Response: We agree that our decision to include the Big Five Inventory was lacking context in the first draft. As we point out in the updated version of our manuscript, including the Items of the Big Five Inventory Short Scale is our attempt to control for potentially unobserved participant characteristics that would systematically distort the influence of social feedback on private opinions. As we are covering new ground with regards to sociological research, we wanted to make sure not to miss out on nuances that could bias our results.

---

## [Editor Report · Decision Letter 1]

7 Sep 2022

The effects of social feedback on private opinions. Empirical evidence from the laboratory.

PONE-D-21-30777R1

Dear Dr. Sarközi,

We’re pleased to inform you that your manuscript has been judged scientifically suitable for publication and will be formally accepted for publication once it meets all outstanding technical requirements.

Kind regards,

Ning Du

Academic Editor

PLOS ONE
---

## [Editor Report · Acceptance letter]

26 Sep 2022

PONE-D-21-30777R1 

The effects of social feedback on private opinions. Empirical evidence from the laboratory. 

Dear Dr. Sarközi:

I'm pleased to inform you that your manuscript has been deemed suitable for publication in PLOS ONE. Congratulations! Your manuscript is now with our production department. 

Kind regards, 

on behalf of

Dr. Ning Du 

Academic Editor

PLOS ONE